# Prediction of Placental Abruption of Pregnant Women Drivers with Various Collision Velocities, Seatbelt Positions and Placental Positions—Analysis with Novel Pregnant Occupant Model

**DOI:** 10.3390/ijerph21070827

**Published:** 2024-06-25

**Authors:** Tomohiro Izumiyama, Atsuno Tsuji, Katsunori Tanaka, Yumiko Tateoka, Ryusuke Asahi, Hiroshi Hamano, Masahito Hitosugi, Shigeru Sugimoto

**Affiliations:** 1Crash Safety Development Department, Vehicle Development Division, Mazda Motor Corporation, Hiroshima 730-8670, Japanasahi.r@mazda.co.jp (R.A.);; 2Department of Legal Medicine, Shiga University of Medical Science, Otsu 520-2192, Japanhitosugi@belle.shiga-med.ac.jp (M.H.); 3Department of Clinical Nursing, Shiga University of Medical Science, Otsu 520-2192, Japan; ytateoka@belle.shiga-med.ac.jp

**Keywords:** placental abruption, collision speed, seatbelt position, placental position, 30-week pregnant occupant, shear, seatbelt loading

## Abstract

The aims of this study were as follows: the (a) creation of a pregnant occupant finite element model based on pregnant uterine data from sonography, (b) development of the evaluation method for placental abruption using this model and (c) analysis of the effects of three factors (collision speed, seatbelt position and placental position) on the severity of placental abruption in simulations of vehicle collisions. The 30-week pregnant occupant model was developed with the uterine model including the placenta, uterine–placental interface, fetus, amniotic fluid and surrounding ligaments. A method for evaluating the severity of placental abruption on this pregnant model was established, and the effects of these factors on the severity of the injury were analyzed. As a result, a higher risk of placental abruption was observed in high collision speeds, seatbelt position over the abdomen and anterior-fundal placenta. Lower collision speeds and seatbelt position on the iliac wings prevented severe placental abruption regardless of placental positions. These results suggested that safe driving and keeping seatbelt position on the iliac wings were essential to decrease the severity of this injury. From the analysis of the mechanism for placental abruption, the following hypothesis was proposed: a shear at adhesive sites between the uterus and placenta due to direct seatbelt loading to the uterus.

## 1. Introduction

In the United States, physical trauma affects 6 to 8% pregnant women and could be the cause of fetal losses during pregnancy [1]. One of the causes of trauma in pregnancy is motor vehicle crashes (MVCs) [2]. Based on the statistical data of MVCs and birth rates, it was estimated that approximately 130,000 women in the second half of pregnancy were involved in MVCs annually in the United States [3]. Additionally, it was reported that placental abruption was a major factor, resulting in negative fetal outcomes up to 3800 fetuses. According to the national population-based study from 1991 to 2001 in Sweden, 207/100,000 pregnant women were involved in MVCs, and 23/100,000 pregnant women sustained maternal severe injuries [4]. In an Australian population-based cohort study, 3.5/100,000 pregnant women experienced MVCs, and the fatality rate of the fetus and neonate was 5.6/100,000 pregnant women [5]. In Japan, a questionnaire survey reported that 2.9% of 2420 women were involved in MVCs during pregnancy [6].

Placental abruption is the partial or complete separation of the uterine–placental interface (UPI) from the uterus before childbirth and accounts for 50 to 70% of negative fetal outcomes in MVCs [7]. The clinical severity of this injury depends on the extent of the separation as follows: under 30%, from 30 to 50%, and over 50%. Those rates are equivalent to fetal fatalities of <30%, 30 to 80% and 80 to 100% after vaginal delivery [8].

Recent findings help to understand the mechanisms of injuries and the kinematics of pregnant occupants in MVCs. Pearlman et al. developed Maternal Anthropomorphic Measurement Apparatus version 2B (MAMA-2B) using a silicone rubber abdomen filled with water [9,10]. They investigated the predictors of fetal loss and the interaction of the dummy with the vehicle interior components in addition to the reproduction of the pregnant kinematics in MVCs. Based on the literature which clarified anthropometric measurements during pregnancy, Rupp et al. improved the anthropometry and the biofidelic response of the abdomen of MAMA-2B [3]. They also implemented design features and instrumentation to assess the predictors of adverse fetal outcomes and discovered the relationship between the uterine inner pressure and the fetal outcome. Motozawa et al. performed sled tests using MAMA-2B to analyze the mechanisms of pregnant injuries in frontal collisions [11]. A sled test is a simplified crash test which makes it possible for engineers to test under various restraint systems in non-destructive crash scenarios. The results indicated that the inner pressures of pregnant drivers were the predictors of fetal outcomes, and a seatbelt was an effective device to protect pregnant drivers. Focusing on the contact between a shoulder belt and the neck in some smaller pregnant women, Hitosugi et al. carried out sled tests using MAMA-2B and confirmed that significant pressures might cause compressions on the neck during frontal collisions due to the inappropriate position of the shoulder belt even if the collision speed was lower [12].

In order to simulate collisions virtually, pregnant finite element (FE) models are also developed. Rupp et al. explored the mechanism of placental abruption using the dummy model and found that this injury was caused due to tensile stresses of more than 15.6 kPa on the UPI or peak uterine strains more than 60% [3]. Thus, two mechanisms were hypothesized: firstly, tensile strain on the UPI due to an anterior–posterior fluid pressure gradient caused by inertial forces and secondly, shear strain or stress on the UPI due to uterus curvature changes caused by direct abdominal loading. Jakobsson et al. made the 36-week pregnant women model including the uterus, placenta, amniotic fluid and fetus to examine the kinematics of the uterus/placenta and the effects of a seatbelt and airbag on fetal outcomes in collisions [13]. It was suggested that pregnant occupants should always wear three-point seatbelts properly, and airbags offered protection for both mothers and fetuses. Moorcroft et al. integrated the 30-week pregnant uterus into the abdomen of the 5th percentile female model using MADYMO (TNO, Netherlands) to investigate how restraint conditions and collision speeds affect the fetal outcome [14]. It was found that the uterine strain was a good predictor of negative fetal outcomes because both the risk of an adverse fetal outcome and the peak uterine strain increased as the collision speed increased in no restraint cases. It was also clarified that the uterine strain depended on the position of the lap-belt and the distance between the steering wheel and the abdomen. Tanaka et al. constructed the FE model of a small 30-week pregnant woman based on Total Human Model for Safety (THUMS) ver1.61 (TOYOTA, Japan) and evaluated placental abruption using the failure criterion of the UPI, 15.6 kPa [15]. It was reported that higher collision speeds and inappropriate uses of a seatbelt increased the area of placental abruption, and a wide lap-belt helped the mitigation of this injury. Kitagawa et al. designed the model of a 30-week pregnant woman based on THUMS ver.1 and concluded that wearing a seatbelt could reduce the loading on pregnant occupants in frontal collisions [16].

It was reported that unbelted pregnant women could have adverse fetal outcomes. From the results of sled tests using MAMA-2B, it was indicated that appropriate uses of a seatbelt during pregnancy could mitigate the severity of placental abruption and prevent negative fetal outcomes [11,17]. Simulations using the pregnant FE model showed that appropriate uses of a seatbelt protected both mothers and fetuses. Tanaka et al. confirmed that a pregnant woman without a seatbelt might put herself and her fetus in danger due to the contact of the steering wheel to the abdomen even if she drove at low speeds [18]. If pregnant drivers fastened with seatbelts on the umbilicus, the area of placental abruption was greater than that of seatbelt position on the iliac wings. Spicka et al. addressed that using a seatbelt over the bump increased the loading on the abdomen and appropriate uses of a seatbelt could significantly reduce it [19].

In the above studies using pregnant occupant models, the placenta was located on the uterine fundus without considering the position of the placenta. Because an interaction between the lap-belt and placenta could be dependent on the position of the placenta, the variabilities in placental distributions should be studied. From the ultrasound imaging of the 30-week pregnant uterus, the placenta was located widely along the wall of the uterus [20]. However, the actual distribution of the placenta might not be reproduced in any previous models. In order to predict the severity of placental abruption during vehicle collisions, it is necessary to create the FE model as a substitute for actual human bodies and develop a method to evaluate the injury based on the model. Therefore, this study has the following aims: the (a) creation of the pregnant occupant model which reproduces an actual distribution of the placenta, based on the ultrasound imaging of the 30-week pregnant uterus, (b) development of evaluation methods for placental abruption using this model and (c) analysis of the effects of three factors, i.e. collision speeds and the position of the seatbelt and placenta, on the severity of placental abruption in simulations of vehicle collisions.

## 2. Materials and Methods

The research outline consists of three steps. The first step is to create the pregnant FE model for the simulations of vehicle collisions. This model was based on THUMS AF05. THUMS is a kind of human body model in simulations as a substitute of actual human bodies, and it has detailed internal organs and defined mechanical properties of human tissues. We selected this model to research occupant kinematics and placental injury. To modify the uterus of THUMS AF05, the functions in pre-processing software Oasys PRIMERTM (V21.0, ARUP, Birmingham, UK) were used. The second step is to establish an evaluation method of placental abruption and validate the biofidelity of the created models using National Automotive Sampling System–Crashworthiness Data System (NASS-CDS) traffic accident data. These data were provided by National Highway Traffic Safety Administration (NHTSA), USA. Finally, the severity of placental abruption was predicted using these models, considering two risk factors (collision speeds and the seatbelt position) and the position of the placenta. Vehicle models including pregnant models were calculated, and the results were analyzed by post-processing software Animater4 (V4, GNS mbH, Braunschweig, Germany).

### 2.1. Creation of Pregnant FE Model

A pregnant occupant model was created based on THUMS AF05 ver.4.02, targeting pregnant women at the equivalent of 30 weeks, when the initial position of a seatbelt and loading location to the uterus are more likely to be changed due to abdominal protrusion, resulting in the considerable effect of the seatbelt path on the severity of placental abruption [14,21]. The uterine model, including the placenta, fetus and UPI layer, was based on the measurement results of each factor in the uterus using ultrasound imaging (Figure 1a) [20]. Furthermore, the distributions of the placenta and amniotic fluid were also reproduced. Smoothed Particle Hydrodynamics (SPH) was applied to the amniotic fluid to reproduce fluid dynamics. The uterus was enlarged to a size equivalent to 30 weeks using the final geometry technique, and the internal organs around the uterus were positioned similar to the actual human anatomy (Figure 1b). Two types of utero-placental models, the anterior- or posterior-fundal placenta, were prepared considering the distribution of placental positions measured in clinical practice (Figure 1c).

### 2.2. Determination of Mechanical Properties for the Uterus, Placenta and UPI Model

The property of the uterus was defined with reference to the S-S properties in tissue tensile tests (Figure 2a) [22]. The properties of the placenta and UPI were defined based on the literature data reported by Hu et al. and Klinich et al. [23,24,25] (Figure 2b,c). Other properties of tissues around the uterus were defined with the literature data reported by Moorcroft et al. [14].

### 2.3. Definition of Seatbelt Position

A survey on the seatbelt usage of 680 pregnant drivers reported that 87% of them used seatbelts appropriately, namely avoiding the abdominal protrusion and positioning on the iliac wings, while 13% used seatbelts in other ways in Shiga Prefecture, Japan [26]. Additionally, the most popular uses of a seatbelt in other ways were to fasten a lap-belt over the abdominal protrusion, and this type accounted for a low rate of pregnant drivers [27]. Considering these data, two types were selected as the method for fastening a seatbelt in FE simulations: a lap-belt position on the iliac wings (proper position, Figure 3a) and over the abdominal protrusion (Figure 3b).

### 2.4. Condition of FE Simulations

In this study, vehicle models used in simulation were based on Mazda’s typical specification, and the 6 DOF accelerations were applied to the vehicle body about its center of gravity (COG) in order to simulate the vehicle kinematics. The acceleration data were obtained from an in-house crash test. The validation results of the simulation model with the Hybrid III AM50 dummy are shown in Figure 4a. The pregnant model was placed in the driver seat and constrained by a seatbelt with a pretensioner and load-limiting retractor without a dynamic locking tongue (Figure 4b). The simulations were conducted with vehicle pulse in a 56 kph and 35 kph full-frontal crash. The models with two types of utero-placental models, the anterior- or posterior-fundal placenta, were used, and the seatbelt was fitted on the chest and lower abdomen or across the abdomen, respectively (Figure 3). Additionally, the occupant was constrained by a knee airbag deployed from the instrumental panel. The seat cushion was squashed to consider the initial compression by gravity. The simplified seating method implemented in Oasys PRIMER^TM^ was used.

### 2.5. The Establishment of an Evaluation Method for Placental Abruption

A method for evaluating the area of placental abruption on the pregnant model was developed. In engineering, strain refers to the deformation such as tension, contraction and twisting that occurs when an external force is applied to an object. Placental abruption is thought to be caused due to localized tensile forces during a seatbelt loading, resulting in the deformation of the uterus and placenta. Therefore, in this study, the threshold for placental abruption in the placenta was defined as the maximum principal strain of 0.58 based on the previous literature [3], and the ratio of the area of placental elements exceeding the value to the total placental elements was calculated. This ratio was compared with the severity of placental abruption used in clinical practice to assess the severity of the injury and fetal outcomes (Figure 5).

### 2.6. Selection of Crash Cases from NASS-CDS Traffic Accident Data

In order to evaluate the biofidelity of the pregnant models by replicating injuries from actual accidents, traffic accident cases were selected from NASS-CDS data [28]. There were 54 cases of pregnant occupant women with third trimester pregnancy and abdominal injuries. Then, 8 cases of pregnant women with uterine or placental injuries were extracted from 54 cases. From the perspective of selecting accidents that are easier to replicate, two cases were ultimately chosen based on the factors of a full-frontal crash or similar condition and pregnant drivers. The validation of the pregnant models with each collision scenario was conducted.

## 3. Results

### 3.1. The Validation of the Pregnant FE Models with NASS-CDS Data

In the first collision scenario (Case ID: 437009969), vehicle V1 was traveling, and the front of V1 struck the front of vehicle V2 that was stopped waiting to make a left-hand turn into an adjoining street. V1 hit V2 at a 10-degree angle at 61 kph. A pregnant woman was the driver of V2, and the airbag was not activated. She was unbelted and suffered the following three injuries: uterus laceration with >50% placental abruption, mesentery laceration and right hip skin abrasion, resulting in fetal death. It was considered that the source of these injuries was the steering wheel. Abdominal injuries were focused on, and both placental abruption and mesentery laceration were reconstructed virtually using the pregnant model. In the study, the area of placental abruption was estimated at about 44%, and high strain was observed in the posterior part of the small intestine, indicating the high possibility of fetal mortality (Figure 6a). These results indicated that the injuries in this MVC were reproduced. In the second accident scenario (Case ID: 777012267), vehicle V1 was stopped in the first lane of traffic. Vehicle V2 was in the same lane and direction. The front of V2 struck the back of V1. A pregnant woman was the driver of V2, and the airbag was activated. She used a seatbelt but over the abdomen and suffered ≤50% placental abruption. Her fetus did not die. In the reconstruction, the weight of the pregnant model was scaled to the occupant weight of 91 kg, and the crash pulse considering the vehicle deformation was used. Because the area of placental abruption was calculated at approximately 26%, this result showed the correlation between the injury in this MVC without fetal mortality and the injury prediction of the pregnant model (Figure 6b).

### 3.2. The Prediction of Placental Abruption—The Effect of Lap-Belt Position

Using the validated pregnant models, the areas of placental abruption were estimated depending on the positions of the placenta and a lap-belt. In the simulations, the positions of the placenta (anterior- or posterior-fundal) and the positions of a lap-belt (on the iliac wings or over the abdomen) were considered. The analysis results are shown in Figure 7. As a result, the abruption area of the pregnant models with a lap-belt position on the iliac wings was less than 10% in each placental position (Figure 8). On the other hand, the abruption area of the models with a lap-belt over the abdomen was predicted at approximately 17% in a model with the anterior-fundal placenta, and the area in that with the posterior-fundal placenta was also higher than the rate of the models with a lap-belt position on the iliac wings although the ratio was less than 10%. These results indicated that the injury risk was higher if a pregnant occupant used a lap-belt over the abdomen.

### 3.3. The Prediction of Placental Abruption—The Effect of Collision Speed

Using the same models, the change in placental abruption in different collision speeds was analyzed. At a collision speed of 56 kph, the simulation results showed that the abruption areas of the models with a lap-belt over the abdomen were wider than those with a lap-belt on the iliac wings and could result in adverse fetal outcomes (Figure 8). At 35 kph, differences in abruption areas between placental positions were also observed although the abruption area was smaller than that at 56 kph. Additionally, it was found that appropriate uses of a seatbelt prevented the fetuses from sustaining negative outcomes because the areas of placental abruption were 1.8 to 9.4% in pregnant models with a lap-belt on the iliac wings. These results indicated that not only collision speed but also maternal restraint uses affected the area of placental abruption in MVCs; furthermore, placental positions were the essential factor to consider the influence of placental abruption.

## 4. Discussion

The current pregnant occupant models have been developed by modifying the previous pregnant model [15] in the following aspects: (1) the use of THUMS AF05 ver4.02 updated to more detailed internal organs and mechanical properties of human tissues, (2) the definition of mechanical properties for the human uterus, placenta, UPI and other tissues around the uterus based on the latest literature and (3) the evaluation of biofidelity by validating the pregnant FE models with actual accident cases extracted from NASS-CDS data. The development of the evaluation method for placental abruption by calculating the maximum principal strain in the placenta of these models has enabled accident reconstruction and the prediction of the severity of placental abruption. As a result, it was found that the anterior-fundal placenta, seatbelt over the abdomen and high collision speed were considered as the factors that resulted in a higher injury risk of placental abruption and adverse fetal outcomes. Moorcroft et al. used the MADYMO pregnant woman model to measure the uterine strain due to three different lap-belt positions and found that the belt position directly on the placenta caused higher strain, and the more upwards position resulted in the lap-belt sliding up and submarining [14]. Spicka et al. studied the effect by five patterns of the lap-belt position and reported that a large loading to the uterus and abdomen was caused in the case of the belt over the abdomen although the belt load was small [19]. Tanaka et al. concluded that the stresses on the placenta were higher for the pregnant occupants with the lap-belt over the abdomen, and the collision speed was a factor to increase the stresses applied to the placenta [15]. The results of our study indicated that the severity of placental abruption increased due to the lap-belt over the abdomen and high collision speed, resulting in negative fetal outcomes, as the same conclusion as previous studies. In addition, it was suggested that differences in placental position affected the severity of this injury. In particular, high collision speeds and the seatbelt over the abdomen were likely to increase the severity of this injury because the anterior-fundal placenta fundamentally promoted high seatbelt loading. On the other hand, the difference between the anterior- and posterior-fundal placenta was small, and there was a low risk of placental abruption in the case of the lap-belt on the iliac wings. In the pregnant models with the posterior-fundal placenta, the phenomenon of placental compression due to a physical interaction between the posterior uterus and lumbar spine was observed after seatbelt loading (Figure 9). Further studies with clinical cases of placental abruption of pregnant women drivers may confirm this phenomenon. However, the severity less than 10% suggested that it was unlikely to result in adverse fetal outcomes. In the future, it is necessary to improve the pregnant model and analysis considering such a phenomenon.

From the analysis using the graphs of Figure 10 and simulations of occupant kinematics, lap-belt loading to the abdomen was observed due to the forward displacement of the pregnant occupant at 60 ms. After this, with a slight delay, the maximum rate of 9.4% placental abruption was confirmed at 70 ms. Based on the above results, one hypothesis was proposed as the mechanism of placental abruption: a shear at adhesive sites between the uterus and placenta due to the local deformation of the uterus resulting from the lap-belt loading to the abdomen (Figure 11). Rupp et al. and Motozawa et al. previously proposed this hypothesis [3,11], and its probability was confirmed in this study. Further verifying this hypothesis would be needed by analyzing the severity of placental abruption on various collision types and speeds, occupant restraints, seat positions and seating postures.

Four limitations in this study are as follows: firstly, only the 30-week pregnant occupants in driver seats were targeted because the initial position of the seatbelt and loading location to the uterus were more likely to be changed due to abdominal protrusion according to the gestational age. Next, the positions of the placenta were selected as two types: the anterior- (marginal placenta previa) and posterior-fundal placenta to the uterus although the other anterior-fundal placentas were found in clinical practice, such as total placenta previa. Then, FE simulations were conducted with vehicle pulse in only a 56 kph or 35 kph full-frontal crash. Finally, only a driver occupant and an upright seating posture were considered.

## 5. Conclusions

In this study, the three following things were achieved. First, novel pregnant FE models with the anterior- and posterior-fundal placenta were created based on THUMS AF05 ver.4.02, targeting pregnant women at the equivalent of 30 weeks. The models were based on the clinical data of the pregnant uterus from sonography. Then, the evaluation method for placental abruption was developed by calculating the ratio of the area of placental elements exceeding the maximum principal strain of 0.58 to the total placental elements in these models. Finally, the effects of three factors, i.e. collision speeds and the position of the seatbelt and placenta, on the severity of placental abruption were analyzed using this method. It was found that the lap-belt position on the iliac wings and lower collision speeds could decrease the risk of this injury regardless of the placental position.

## Figures and Tables

**Figure 1 ijerph-21-00827-f001:**
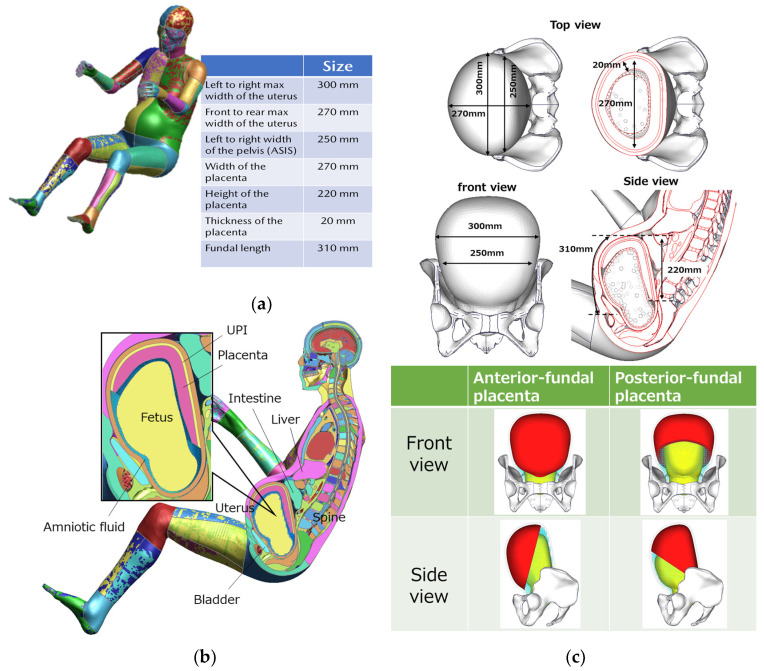
(**a**) The measurement data of the uterus and placenta; (**b**) the pregnant occupant model; (**c**) the position of the placenta in the uterus.

**Figure 2 ijerph-21-00827-f002:**
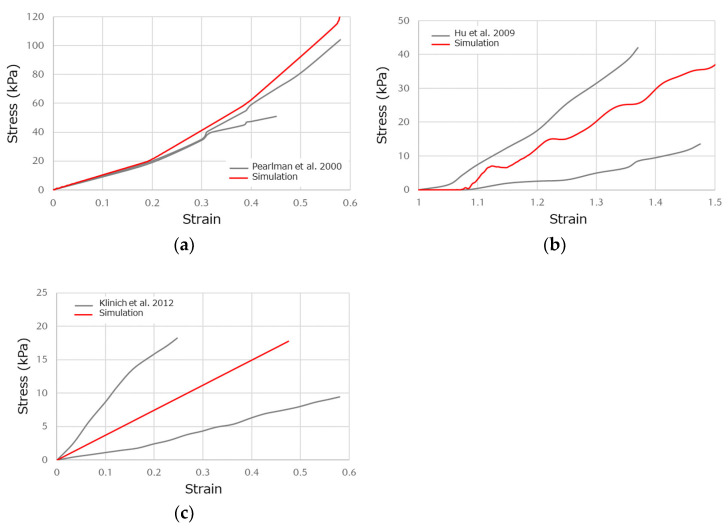
The S-S properties in tissue tensile tests: (**a**) the uterus; (**b**) the placenta; (**c**) the UPI.

**Figure 3 ijerph-21-00827-f003:**
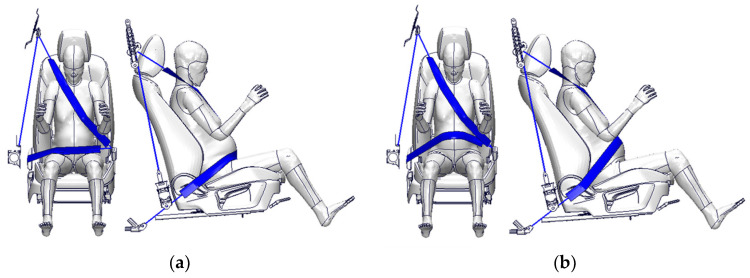
The initial lap-belt position: (**a**) on the iliac wings and (**b**) over the abdomen.

**Figure 4 ijerph-21-00827-f004:**
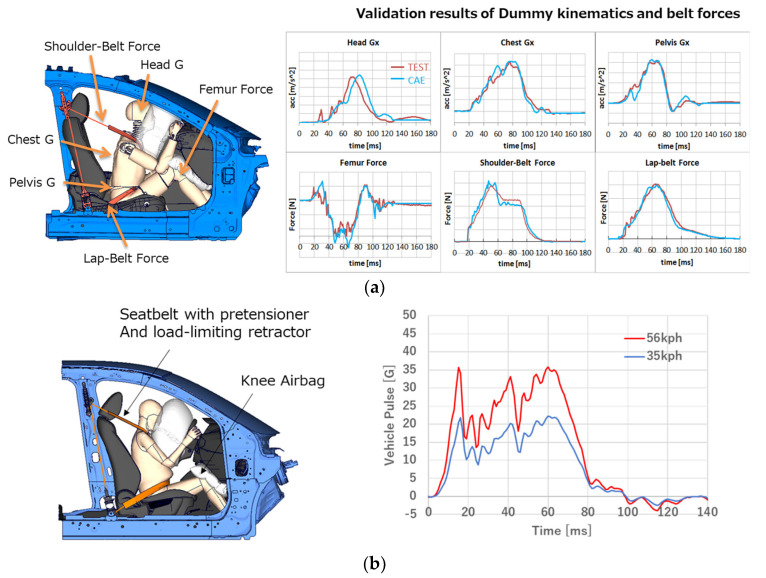
(**a**) The validation results of the vehicle model; (**b**) the condition of FE simulations using pregnant models.

**Figure 5 ijerph-21-00827-f005:**
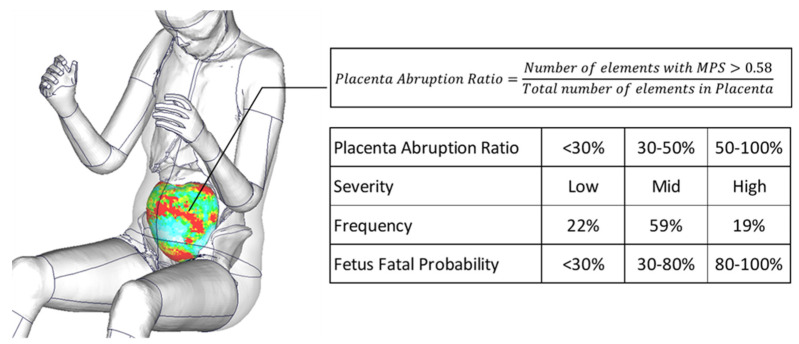
The evaluation method for the severity of placental abruption and fetal outcomes.

**Figure 6 ijerph-21-00827-f006:**
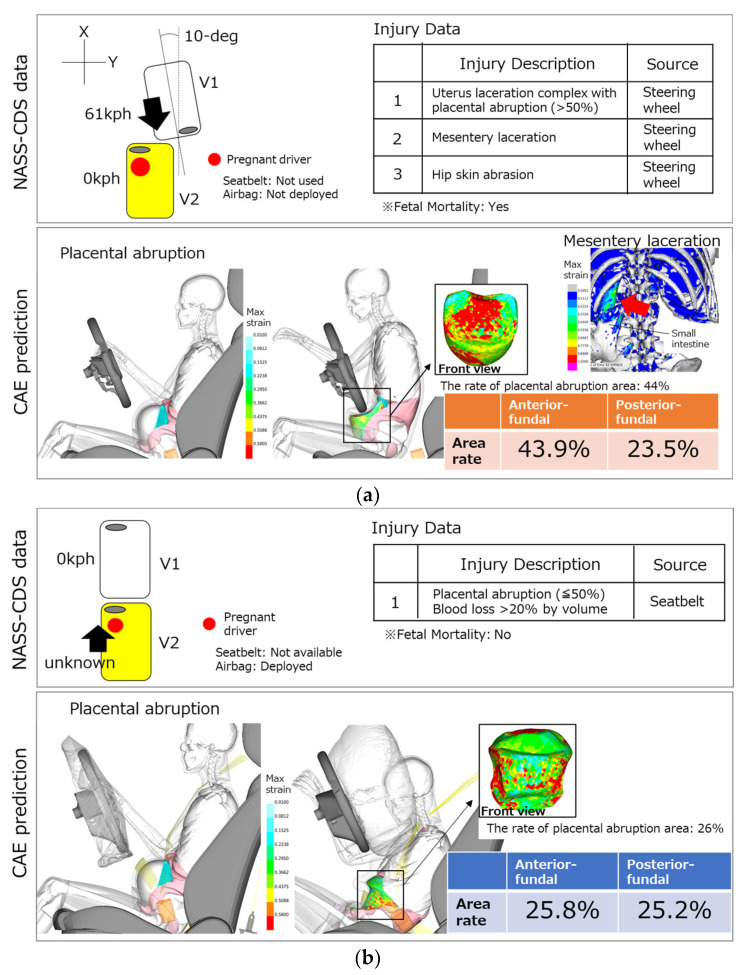
The validation results of the pregnant models with NASS-CDS: (**a**) Case ID 437009969; (**b**) Case ID 777012267.

**Figure 7 ijerph-21-00827-f007:**
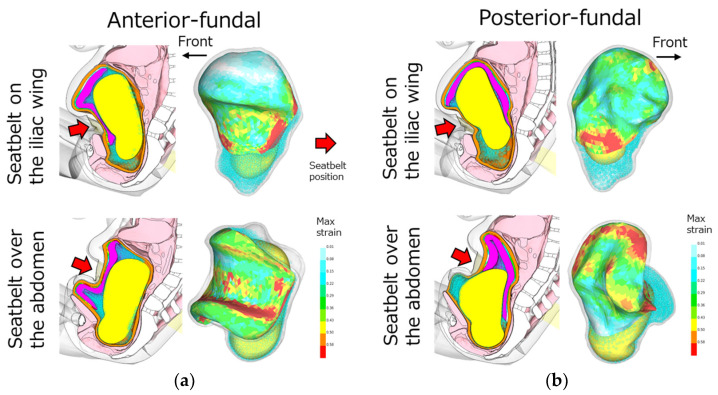
The analysis results of the lap-belt positions (on the iliac wings or over the abdomen) for the placental positions: (**a**) anterior-fundal placenta and (**b**) posterior-fundal placenta. Red arrows show the position of the seatbelt loading during a collision.

**Figure 8 ijerph-21-00827-f008:**
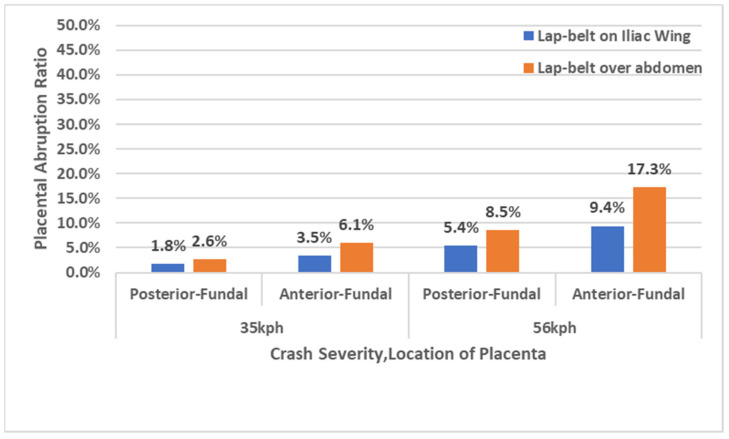
The ratio of placental abruption with a lap-belt position on the iliac wings or over the abdomen in collision speeds of 35 kph and 56 kph.

**Figure 9 ijerph-21-00827-f009:**
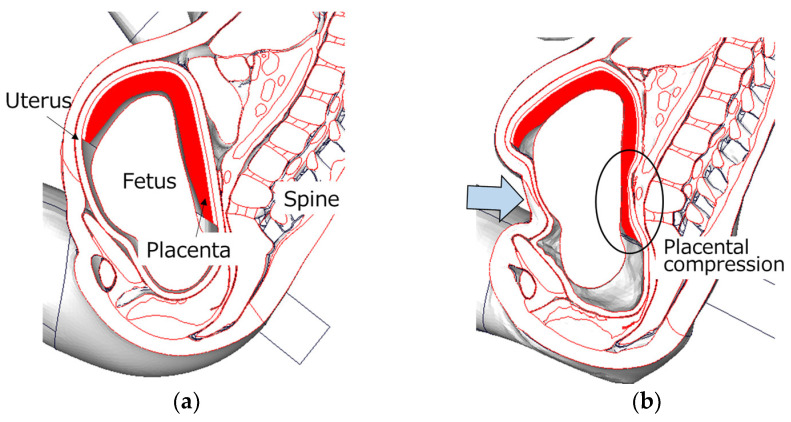
The state of the placenta in a pregnant occupant with a posterior-fundal placenta: (**a**) at the initial time and (**b**) at the time of seatbelt loading. The light blue arrow shows the position of the seatbelt loading during a collision.

**Figure 10 ijerph-21-00827-f010:**
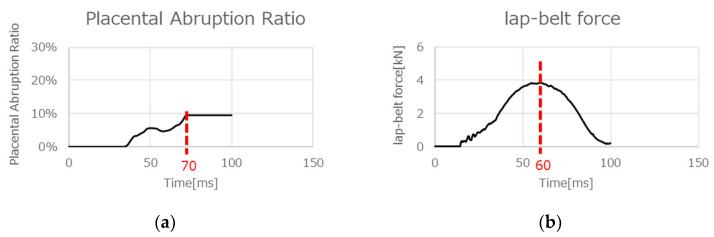
The analysis results with an anterior-fundal placenta in a pulse of 56 kph: (**a**) lap-belt force and (**b**) placental abruption rate.

**Figure 11 ijerph-21-00827-f011:**
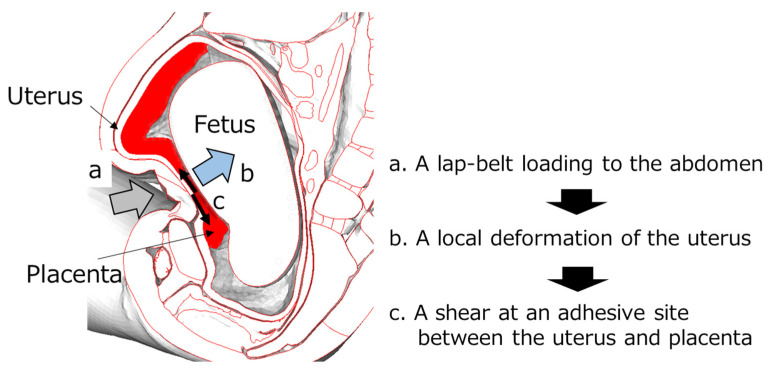
The hypothesis of the mechanism of placental abruption: a shear between the uterus and placenta due to the seatbelt loading.

## Data Availability

The pregnant occupant models used in this study themselves cannot be shared without a contract, such as collaborative research, unless the THUMS USER POLICY is satisfied. However, the data that support the findings of this study can be made available by the authors upon reasonable request.

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
