# Peer review of "Prediction of Placental Abruption of Pregnant Women Drivers with Various Collision Velocities, Seatbelt Positions and Placental Positions—Analysis with Novel Pregnant Occupant Model"

_ijerph, 2024, doi:10.3390/ijerph21070827_

Round 1

Reviewer 1 Report

Comments and Suggestions for Authors

Dear authors, it is a great pleasure to review your study entitled”  Prediction of placental abruption of pregnant women drivers with various collision velocity, seatbelt position and placental positions -Analysis with the novel pregnant occupant model”. If I am not wrong an article has been published by authors of this manuscript two years ago (PMID: 36360785 PMCID: PMC9656600). If I am right, then what is the reason for doing or reporting of this study?

Please justify the need or novelty of this article.

Abstract:

What is FE model? UPI?

“(a) creation of pregnant occupant FE model based on pregnant uterine data from sonography, (b) development of the evaluation method for placental, abruption using this model and (c) analysis of the effects of three factors (collision speed, seatbelt position and placental position) on the severity of this injury”

Severity of which injury?

Why was considered a, b, c phases in this study?

What is kind of study?

When was this study done?

What was inclusion criteria?

What was the outcomes of study?

“The 30-week pregnant occupant model” why was chosen 30week pregnant women?

“A method was established to evaluate the severity of placental abruption on this pregnant model, and the effects of these factors on the severity of the injury were analyzed.” What method?

How was this study performed?

What risk factors were considered for prediction of models?

What is THUMS?

Materials and Methods:

What is FE-HBM?

What kind is this study? Simulation?

Why is strain and stress was reported?

The quality of figures are not appropriate. For example, figure 8: arrows what do show?

Author Response

Thank you for your thoughtful and constructive feedback on our manuscript. We are grateful for the time and energy you have spent on our behalf. We have revised the manuscript based on your questions. Additionally, we would like to answer your questions as follows:

Dear authors, it is a great pleasure to review your study entitled ”Prediction of placental abruption of pregnant women drivers with various collision velocity, seatbelt position and placental positions -Analysis with the novel pregnant occupant model”. If I am not wrong an article has been published by authors of this manuscript two years ago (PMID: 36360785 PMCID: PMC9656600). If I am right, then what is the reason for doing or reporting of this study?

Please justify the need or novelty of this article.

The pregnant finite element model created based on THUMS ver1 was used in the manuscript published two years ago (PMID: 36360785 PMCID: PMC9656600). However, we created the current pregnant models based on THUMS ver4 updated to more detailed internal organs and mechanical properties of tissues in this study. Additionally, we defined new mechanical properties of the human uterus, placenta, uterine-placental interface and other tissues around the uterus to these models with reference to the latest literatures. Furthermore, these models were validated with actual accident cases to evaluate the biofidelity of the models and actual injuries were replicated. In these respects, the models used in this study differ from previous pregnant model (PMID: 36360785 PMCID: PMC9656600) and our models are closer to actual pregnant women. We have included these contents in the first part of Discussion section. Please refer to the revised paper.

Abstract:

What is FE model? UPI?

FE model stands for finite element model and is used in simulations as a substitute for actual human bodies, vehicles, and so on. UPI is the uterine-placental interface between the uterus and the placenta. Please refer to lines 15 and 19-20 in a revised paper.

“(a) creation of pregnant occupant FE model based on pregnant uterine data from sonography, (b) development of the evaluation method for placental abruption using this model and (c) analysis of the effects of three factors (collision speed, seatbelt position and placental position) on the severity of this injury”

Severity of which injury?

“This injury” means placental abruption. We revised this part in abstract section. Please refer to line 18 in the revised paper.

Why was considered a, b, c phases in this study?

In order to predict the severity of placental abruption during vehicle collisions, it is necessary to create the FE model as a substitute for actual human bodies and develop a method to evaluate the injury based on the model.

What is kind of study?

This study is kind of simulation using the FE model.

When was this study done?

This study was done after the study performed by Tanaka et al. (PMID: 36360785 PMCID: PMC9656600).

What was inclusion criteria?

No experiments involving subjects were conducted in this study. This study only involves simulations using the FE models.

What was the outcomes of study?

As mentioned in the initial response, the following things are the outcomes of this study:

  • Creation of the pregnant FE model as a substitute for actual human bodies
  • Development of the evaluation method for placental abruption using this model
  • Prediction of the severity of placental abruption considering three factors, vehicle collision speeds, seatbelt positions and placental positions

“The 30-week pregnant occupant model” why was chosen 30week pregnant women?

By around the 30th week, the effect of the seatbelt path on the severity of placental abruption becomes significant due to the abdominal protrusion.

“A method was established to evaluate the severity of placental abruption on this pregnant model, and the effects of these factors on the severity of the injury were analyzed.” What method?

I apologize for writing confusing English. We revised “A method for evaluating the severity of placental abruption on this pregnant model was established”. Please refer to lines 20-21 in Abstract section and lines 171-172 in Materials and Methods section 2.5 of the revised paper.

How was this study performed?

Simulations using the pregnant FE models were analyzed and predicted the severity of placental abruption by evaluating strains on the placenta.

What risk factors were considered for prediction of models?

Risk factors are collision speeds and the position of seatbelt.

What is THUMS?

THUMS stands for Total Human Model for Safety and was created by TOYOTA, Japan. This is a kind of human body models in simulations as a substitute for actual human bodies. THUMS has detailed internal organs and define mechanical properties of human tissues. Other models are GHBMC, SAFER-HBM, and so on.

Materials and Methods:

What is FE-HBM?

HBM stands for human body model. We changed “HBM” to “model” to avoid confusion. Please refer to line 118 in Materials and Methods section 2.1 of the revised paper.

What kind is this study? Simulation?

This study is kind of simulation using the FE model.

Why is strain and stress was reported?

In engineering, strain refers to the deformation such as tension, contraction, and twisting that occurs when an external force is applied to an object. We believe that placental abruption occurs due to localized tensile forces during a seatbelt loading, resulting in deformation of the uterus and placenta. Therefore, placental abruption was evaluated using strain in this study. Actually, a similar assessment has been conducted in previous study [3].

The quality of figures are not appropriate. For example, figure 8: arrows what do show?

We added an explanatory sentence in each figure caption. Please refer to lines 227-228 in figure 8 and lines 281-282 in figure 10 of the revised paper.

Reviewer 2 Report

Comments and Suggestions for Authors

Dear Authors

The topic is interesting and the manuscript was written well.

-To make the study more practical please explain how can we use the result of this study in clinical practice?

-How do you evaluate the effect of uterine scars in pregnant woman with a previous history of myomectomy or cesarean section in the calculation of strain at the uterus in trauma?

-please explain the technical parts of the method section clearly.

Author Response

Thank you for your thoughtful and constructive feedback on our manuscript. We are grateful for the time and energy you have spent on our behalf. We would like to answer your questions as follows:

-To make the study more practical please explain how can we use the result of this study in clinical practice?

Thank you so much for your kind and thoughtful consideration. We would like to make the following information known not only to pregnant women but also to obstetricians:

  • Appropriate position of the seatbelt (the lap-belt position on the iliac wings) is very important, whereas improper position, such as the lap-belt position over the abdominal protrusion, could increase the risk of placental abruption. We believe these results will be useful for educating pregnant women.
  • Pregnant women with the anterior-fundal placenta are susceptible to severe placental abruption in the event of particularly traffic accidents. Therefore, doctors should encourage them to seek medical attention even after minor collisions.

We have also been conducting awareness activities regarding seatbelt usage for pregnant women since last year. We believe that more pregnant women will learn the appropriate usage of the seatbelt through this activity.

-How do you evaluate the effect of uterine scars in pregnant woman with a previous history of myomectomy or cesarean section in the calculation of strain at the uterus in trauma?

Mechanical properties of women with a history of uterine surgery have not been clarified. This study assumes that the pregnant women are healthy individuals.

-please explain the technical parts of the method section clearly.

The points of technical parts are as follows:

(1) Creation of the pregnant FE model considering the distribution of placental positions based on clinical data of pregnant uterus from sonography

(2) Definition of mechanical properties for the actual human uterus, placenta, UPI and other tissues around the uterus to the model

(3) Development of the evaluation method for placental abruption by the strain on the placenta of the pregnant FE model

Reviewer 3 Report

Comments and Suggestions for Authors

Thank you for allowing me to review this very interesting manuscript.

Abstract: Suggest to authors that they write out FE and UPI abbreviations as noted in Abstract lines 15 and 19.

Introduction:  It is not clear whether the authors' first sentence is in reference to the US or globally, would consider indicating the scale and/or place.  

Line 60 is the first mention of sled test.  Suggest to the authors that a brief description of this test and its significance in the overall testing mechanisms be included.

Line 75-delete "of"; line 133 add "the" before uterine and suggest using uterus instead.

Materials & Methods

Suggest to the authors that they indicate where the study was performed.  It is unclear until the 2.3 section, line 141 that it is in Japan.

The methods section 2.3 seems to indicate that there are two types of studies here.  Survey with real subjects and a testing of models.   Suggest to authors that they make this clear and indicate if IRB and informed consent was obtained and change lines 315-316. Survey details, participant recruitment, and other original research data information need to be indicated.  Otherwise suggest taking out section 2.3 for separate paper

figure 4 lines 166-168, suggest adding appropriate explanation. 

Suggest to authors that the introduction of the use of NASS-CDS data set be introduced before the results section.

Results 

line 185 suggests deleting "was" before hit.

In section 3.1 in the retelling of the MVC case, it is not clear initially whether there was fetal mortality, such to authors that they include fetal outcome around line 188. I found this out by Figure 7 case a.  

Line 199-there is a reference to a bonnet, not sure where that comes from don't see that early in methods, suggest if you leave it in then go back and explain clearly the use of the bonnet and what it tells the readers.  

Section 3.2 it would be helpful if authors included relevant NASS-CDS Traffic Accident Date cases for the placental position results.  

Discussion 

Is is unclear to me if the answer to the second aim of the study was met, suggest making that clear to readers in the discussion or results.  

Will the deployment of airbags during a crash make a difference in the outcomes and incidence of placenta abruption?  It is clear that seatbelts clearly make a difference.  

Thank you for allowing me to read such an interesting and engaging article. 

Comments on the Quality of English Language

There were minor English revisions needed in present state.  

Author Response

Thank you for your thoughtful and constructive feedback on our manuscript. We are grateful for the time and energy you have spent on our behalf. We have revised the manuscript in accordance with your suggestions. Please confirm as follows:

Abstract: Suggest to authors that they write out FE and UPI abbreviations as noted in Abstract lines 15 and 19.

Following your suggestion, we modified FE and UPI abbreviations. Please refer to lines 15 and 19 in a revised paper.

Introduction:  It is not clear whether the authors' first sentence is in reference to the US or globally, would consider indicating the scale and/or place.  

We added “In the United States” in the line 33 of a revised paper.

Line 60 is the first mention of sled test.  Suggest to the authors that a brief description of this test and its significance in the overall testing mechanisms be included.

Following your suggestion, we added a sentence about sled test. Please refer to lines 61-62 “A sled test is …” in a revised paper.

Line 75-delete "of"; line 133 add "the" before uterine and suggest using uterus instead.

We modified according to your suggestions. Please refer to lines 76 and 134 in a revised paper.

Materials & Methods

Suggest to the authors that they indicate where the study was performed. It is unclear until the 2.3 section, line 141 that it is in Japan.

We modified the sentence to indicate where the study was conducted. Please refer to lines 142-144 in a revised paper.

The methods section 2.3 seems to indicate that there are two types of studies here. Survey with real subjects and a testing of models. Suggest to authors that they make this clear and indicate if IRB and informed consent was obtained and change lines 315-316. Survey details, participant recruitment, and other original research data information need to be indicated. Otherwise suggest taking out section 2.3 for separate paper.

Surveys with real subjects were conducted previously by Hanahara et al. [25] and Acar et al. [26]. In this study, we tested only FE simulations using models with two types of seatbelt positions. The positions were based on the results of above surveys. We have changed the sentence of citing reference [25] to make that clear. Please refer to section 2.3 in the revised paper.

figure 4 lines 166-168, suggest adding appropriate explanation. 

Following your suggestion, we revised the caption of figure 4. Please refer to the lines 167-168 in the revised paper.

Suggest to authors that the introduction of the use of NASS-CDS data set be introduced before the results section.

For your suggestion, we added the introduction regarding the use of NASS-CDS data set in Materials and Methods section 2.6. Please refer to the lines 178-186 of the revised paper.

Results 

line 185 suggests deleting "was" before hit.

Following your suggestion, we deleted “was” before hit in line 191 of the revised paper.

In section 3.1 in the retelling of the MVC case, it is not clear initially whether there was fetal mortality, such to authors that they include fetal outcome around line 188. I found this out by Figure 7 case a.  

We added a sentence regarding fetal mortality in lines 194 and 198-199 of the revised paper. Please confirm.

Line 199-there is a reference to a bonnet, not sure where that comes from don't see that early in methods, suggest if you leave it in then go back and explain clearly the use of the bonnet and what it tells the readers.  

We revised the sentence to a clearer expression. Please refer to lines 205-206 in the revised paper.

Section 3.2 it would be helpful if authors included relevant NASS-CDS Traffic Accident Date cases for the placental position results.  

We also agree with your suggestion. However, the information regarding the placental position is not included in NASS-CDS data. Therefore, we analyzed the rates of placental abruption considering two cases (both anterior and posterior attached placenta) and indicated the probability of the placental position in each case.

Discussion 

It is unclear to me if the answer to the second aim of the study was met, suggest making that clear to readers in the discussion or results.  

Following your suggestion, we added a sentence about an achievement of the second aim of the study. Please refer to lines 250-252 in the discussion of the revised paper.

Will the deployment of airbags during a crash make a difference in the outcomes and incidence of placenta abruption?  It is clear that seatbelts clearly make a difference. 

We hypothesize that airbags dose not affect the severity of placental abruption because airbags deploy in different positions from the uterus and have almost no interaction with the uterus, while a seatbelt directly loads the uterus.

Round 2

Reviewer 1 Report

Comments and Suggestions for Authors

Dear Authors, Many thanks for your efforts in revising the manuscript. Please apply all my previous questions to the manuscript as well. You have worked hard and answered the questions in a separate sheet, please apply all previous questions and current questions in the manuscript file itself.

This study was a simulated study in the laboratory ??? To check the safety of these results suggested that safe driving and keeping seatbelt position on the iliac wings were essential to decrease the severity of this injury has been done. With what software was this simulation done? What was the work process? After the detailed explanations, the authors should explain about the models. And it should be explained why these models were chosen?

It has been written that "By around the 30th week, the effect of the seatbelt path on the severity of placental abruption becomes significant due to the abdominal protrusion". Based on what reference?

Author Response

Comment 1: Dear Authors, Many thanks for your efforts in revising the manuscript. Please apply all my previous questions to the manuscript as well. You have worked hard and answered the questions in a separate sheet, please apply all previous questions and current questions in the manuscript file itself.

Response 1: Thank you for kindly reviewing our manuscript again. I’m sorry for not reflecting some answers in our manuscript. According to your suggestion, we modified it with all answers including previous answers. They are indicated by red words in the revised paper. Additionally, we have indicated the passages of the manuscript where we included our previous responses in red for each of the following Round1 questions. Please confirm them.

Comment 2: This study was a simulated study in the laboratory ??? To check the safety of these results suggested that safe driving and keeping seatbelt position on the iliac wings were essential to decrease the severity of this injury has been done. With what software was this simulation done? What was the work process? After the detailed explanations, the authors should explain about the models. And it should be explained why these models were chosen?

Response 2: Following your suggestion, we added the outline of this study including simulation software, the process, and so on in the first part of Materials and Methods section. Please refer to the revised paper.

Comment 3: It has been written that "By around the 30th week, the effect of the seatbelt path on the severity of placental abruption becomes significant due to the abdominal protrusion". Based on what reference?

Response 3: Following your suggestion, we added the reason for a selection of the 30-week pregnant women and the references in lines 142-143 of Materials and Methods section 2.1. Please refer to the revised paper.

Reviewer 3 Report

Comments and Suggestions for Authors

thank you for your thorough submission and suggested corrections.  It is a fascinating paper.  Thank you for the opportunity to review.  

Author Response

Comment 1: Thank you for your thorough submission and suggested corrections.  It is a fascinating paper.  Thank you for the opportunity to review. 

Response 1: Thank you for kindly reviewing our revised manuscript again.